# Effect of Clay Nanofillers on the Mechanical and Water Vapor Permeability Properties of Xylan–Alginate Films

**DOI:** 10.3390/polym12102279

**Published:** 2020-10-04

**Authors:** Darrel S. Naidu, Maya J. John

**Affiliations:** 1CSIR, Centre for Nanostructures and Advanced Materials, Pretoria 0184, South Africa; mjohn@csir.co.za; 2Department of Chemistry, Nelson Mandela University, Port Elizabeth 6031, South Africa

**Keywords:** food packaging, biopolymer, nanocomposite, edible films, water sorption, water vapor permeability

## Abstract

In this study, xylan–alginate-based films were reinforced with nanoclays (bentonite or halloysite) by the solvent casting technique. The effect of the nanoclay loadings (1–5 wt %) on various properties—mechanical, optical, thermal, solubility, water sorption, and water vapor permeability (WVP)—of the xylan–alginate films were examined for their application as food packaging materials. A 5 wt % loading of either bentonite or halloysite resulted in a 49% decrease of the WVP due to the impermeable nature of the silicate layers that make up both bentonite and halloysite. Thermal stability and solubility of the nanocomposite films were not significantly influenced by the presence of the nanoclays, whereas the optical properties were significantly improved when compared to neat xylan–alginate blend. In general, films reinforced with bentonite exhibited superior mechanical and optical properties when compared to both halloysite-based nanocomposite and neat films.

## 1. Introduction

Polymers derived from petroleum-based resources are increasingly used as packaging materials due to their low cost and favorable packaging properties. However, the rising environmental concerns regarding the non-biodegradable nature of synthetic plastics and the resultant waste pollution has stimulated the need for viable bio-based alternatives. Presently, there is a great demand for biopolymer-based packaging materials due to the advantages of being renewable, biodegradable, biocompatible, having a limited environmental impact, and absence of end-of life disposal issues. Biopolymers such as chitosan, pectin, gelatin, and starch are being developed for food packaging applications [1].

Xylan and alginate represent two polysaccharides that are good candidates to replace the petroleum-based single-use film plastics that are currently being used for food packaging applications. Both biopolymers can be extracted from various agricultural waste residues (maize stalks, sugarcane bagasse, wheat straw, etc.) and marine resources (seaweed), which are available abundantly and are currently underutilized [2,3,4,5]. From a South African perspective, agricultural waste residues such as maize stalks and bagasse residues are usually burnt or dumped in the field, without any meaningful conversion or value-addition.

Xylan is the most common form of hemicellulose which is a class of heteropolymers consisting of pentose and hexose sugars. Xylan films exhibit properties that are suitable for food packaging applications such as formation of transparent films, low oxygen permeability, and strong barrier to oils and fats [6]. However, xylan has poor film forming properties and needs to be blended with plasticizers or other biopolymers to form films. Alginate has good film forming properties and the films have properties suitable for food packaging applications such as high resistance to fats and oils [7]. Alginate films, however, are brittle and yellow in color which limit their application as food packaging materials. However, both xylan and alginate suffer from two main drawbacks, namely, poor mechanical properties and a highly hygroscopic nature [7,8,9,10,11]. Nanofillers which are abundantly available can be incorporated to improve mechanical and water-resistant properties of the films. A number of studies describe the use of different types of nanofillers (graphene oxide, nanocellulose, titanium dioxide, and silica) in both xylan and alginate to improve their properties [11,12,13,14].

Nanoclays are a class of reinforcement material that are both abundantly available and have dimensions in the nanoscale. Nanoclays are made up of layered silicates that are held together by Van der Waals forces. The layered silicates can be joined to either aluminum hydroxide or magnesium hydroxide [15,16]. Bentonite is a type of nanoclay that takes the form of platelets and consists of alternating layers of silica and aluminum hydroxide sheets in a 2:1 ratio [17]. Bentonite has applications in the biomedical field as a food additive and in the pharmaceutical industry as an excipient [18,19]. Several studies have shown that bentonite improves the properties of biopolymer films for food packaging applications [20,21,22]. Halloysite is another type of nanoclay, with a morphology similar to that of carbon nanotubes [23,24]. Halloysite consists of silica and aluminum hydroxide sheets in a 1:1 ratio [25]. Halloysite is used in the ceramics industry due to its suitable thermal properties and in the pharmaceutical industry as it can release drugs in a controlled manner [26]. A number of studies have reported that the incorporation of halloysite improves the mechanical and barrier properties of biopolymer films [27,28,29].

This study reports on the preparation and characterization of xylan–alginate films that were reinforced with either bentonite or halloysite (in varying ratios) to improve properties for food packaging applications. The films were prepared by the solvent casting technique and the barrier, mechanical, morphological, and thermal properties of the films were assessed. The effects of bentonite and halloysite on the film properties were compared to determine which clay acted as a better reinforcement material. To the best of our knowledge, there are no studies done on this combination of xylan–alginate blends with bentonite or halloysite as a reinforcement material. Furthermore, there are no known studies available that compare bentonite and halloysite as reinforcement materials for food packaging applications.

## 2. Materials and Methods

### 2.1. Materials

Xylan (with a purity of 95%) was purchased from Leapchem, China. Sodium alginate, glycerol (with a purity of 99.5%), calcium chloride, bentonite, and halloysite were procured from Sigma Aldrich, South Africa. All chemicals were used without further purification.

#### Preparation of Xylan–Alginate Films with Clay Fillers

The xylan blends were prepared using the solvent casting method. Xylan blends were produced by combining 3.6 g of alginate, 3.6 g of xylan, 0.8 g of glycerol, and a weighed amount of clay (given in Table 1) in a 250 mL beaker. The mixture was then dissolved in 250 mL of deionized water by stirring for 8 h at 500 RPM using a mechanical stirrer. The solution was then sonicated for 2 h and transferred to a tray (23 cm × 22 cm). The trays were then placed in a fan oven at 50 °C for 16 h to dry the samples. The samples were removed from the trays and then stored in a chamber at 54% constant humidity and 23 °C until analysis.

### 2.2. Characterization

#### 2.2.1. Fourier Transform Infrared Spectroscopy (FTIR)

The FTIR spectra were obtained using a PerkinElmer Spectrum 100, FTIR spectrometer. The scan range was 650 to 4000 cm^−1^, with a scan resolution of 4 cm^−1^. An average of 16 scans was used.

#### 2.2.2. X-ray Diffraction

X-ray diffraction (XRD) patterns of bentonite, halloysite, and xylan–alginate films containing bentonite and halloysite were obtained using a Bruker D2 Phaser benchtop XRD, with a 2°/min step, λ = 0.15433 nm, 40 kV, and a scan range of 5–80°.

#### 2.2.3. Colorimetric Analysis

A Konica Minolta CM-2600d was used to measure the color parameters of the blends, using the CIE L*, a*, and b* color scale. The value L* represents the darkness of the sample, with 100 being absolute white and 0 being absolute black [30]. The value a* represents the redness or greenness, with the values ranging from +60 to −60 and the value +60 is red and the value −60 is green. The value b* represents yellow and blue with the values ranging from +60 to −60, with +60 being yellow and −60 being blue. The readings were carried out at room temperature and an average of five readings were taken from various parts of the film. A sheet of white paper was used as control to verify the suitability of the films towards packaging, given that a close match indicates the resulting color and its intensity.

#### 2.2.4. Light Transmittance

The light transmittance spectra of the film blends were obtained using a PerkinElmer Lambda 35. The samples were cut into 2 cm × 3 cm strips and placed into the solid sample holder of the instrument. Air was used as the blank, a scan rate of 240 nm/min was used, and the range of the scan was from 200 to 800 nm. An average of 5 samples was used for each film blend.

#### 2.2.5. Mechanical Properties

Tensile properties of the films were measured on an Instron 3345 tensile tester equipped with a 50 N load cell and a gauge length of 50 mm according to ASTM D882 standards. The rectangular strips with dimensions of 100 mm × 10 mm (length × breadth) were cut from the films and the tensile measurements were performed at a strain rate of 25 mm min^−1^. Ten specimens were tested for every blend sample, and the mean and standard deviation were reported.

#### 2.2.6. Thermogravimetric Analysis

Thermogravimetric analysis (TGA) was recorded using a TA instruments Discovery series Hi-Res thermogravimetric analyzer. The analysis was carried out using the dynamic mode, resolution setting of 4, and a sensitivity setting of 1. The analysis was carried out under nitrogen (99.99% purity) flow at a flow rate of 25 mL min^−1^. The sample mass was ~10 mg and the analysis was carried out from 25 to 450 °C.

#### 2.2.7. Water Vapor Permeability

Water vapor permeability (WVP) measurements were done according to ASTM E96-05 (ASTM, 2005). The films were sealed onto aluminum cups that contained ~42 g of CaCl_2_ and had an air gap of approximately 6 mm between the desiccant and the film. The cups were then placed into a desiccator cabinet, which contained a solution of saturated NaCl to provide an atmosphere of 75% relative humidity. Air in the cabinet was circulated using a fan, and the temperature of the cabinet was maintained at 23 °C.

Before each weighing, the temperature and humidity of the cabinet were recorded, using a Rotronic HygroPalm RH meter. The water vapor transmission rate (WVTR) was calculated from Equation (1).
(1)WVTR=ΔmtA,
where Δmt is the slope of the straight line of the graph plotted of weight gained vs. time and *A* is area of the cup mouth.

The WVP was calculated from Equation (2).
(2)$$WVP=\frac{WVTR\times L}{\Delta p},$$
where L is the thickness of the film and Δp is the partial pressure between the two sides of the film. An average of three films was tested and the mean and standard deviation was reported.

#### 2.2.8. Water Sorption Isotherms

Water sorption isotherms were obtained by cutting the dried films into strips and weighing out ~0.250 g of each film into aluminum foil cups. These cups were then placed into a humidity chamber with various saturated salt solutions, [NaOH, MgCl_2_, MgNO_3_·6H_2_O, NaCl, and K_2_SO_4_] to achieve relative humidity values of 11%, 33%, 54%, 75%, and 96%, respectively [31,32]. The cups were measured every 24 h for 5–7 days until the sample weight was within 0.001 g for 2 consecutive days. The temperature of the chamber was maintained at 23 °C and the measurements were done in triplicate. Moisture content was calculated according to Equation (3).
(3)$$MC\%=\frac{W_{RH}-W_{I}}{W_{I}}\times 100\%,$$
where MC% is the percentage of moisture absorbed relative to the dry mass of the film. $W_{RH}$ is the weight of the film at a given relative humidity and $W_{I}$ is the dry mass of the film.

The water sorption isotherms were analyzed using the Guggenheim–Anderson–de Boer (GAB) model in Equation (4) [33].
(4)$$MC=\frac{W_{m}\cdot C\cdot K\cdot a_{w}}{\left(1-K\cdot a_{w}\right)\left(1-K\cdot C\cdot a_{w}+K\cdot a_{w}\right)},$$
where *MC* is the percentage moisture content, and *a_w_* is the water activity (relative humidity). The terms *K*, *C*, and *W_m_* are model constants. A graph of $\frac{a_{w}}{MC}$ vs. *a_w_* is plotted, which takes the form of a 2nd order polynomial ($y=ax^{2}+bx+c$). The constants *K*, *C*, and *W_m_* are calculated from the resulting graph by using Equations (5)–(7), respectively [33].
(5)$$aK^{2}+bK+c=0,$$
(6)$$C=\frac{b}{aK}+2,$$
(7)$$W_{m}=\frac{1}{b+2Ka}.$$

The constants *a*, *b*, and *c* are obtained from a plot of $\frac{a_{w}}{MC}$ vs. *a_w_*.

#### 2.2.9. Solubility Studies

Films were cut into strips and dried overnight at 105 °C. The dried film weighing ~0.2 g was placed in a beaker containing 20 mL of water and the mixture was stirred using a magnetic stirrer at 60 rpm for 24 h. Samples were then filtered through a Gooch crucible (porosity 4) that had been previously weighed. The crucibles were dried overnight at 105 °C and reweighed, and the difference in weight was taken as the insoluble residue of the films. An average of three replicates was used.

#### 2.2.10. Scanning Electron Microscopy

Scanning electron microscopy (SEM) was used to observe the surface and cross-section microstructure of the xylan–alginate films. The films were cut into 0.5 cm × 0.5 cm strips, before being placed onto a stub holder using carbon tape, and were sputter-coated with gold using an Emitech K575x sputter coater. The cross-section of the films was taken by fracturing the films, which were dried overnight in an oven at 105 °C. The images were taken using a JOEL JSM-7001F at an acceleration voltage of 5 kV.

## 3. Results and Discussion

### 3.1. FTIR

Figure 1 shows the FTIR spectra of xylan, alginate, bentonite, halloysite, and films containing bentonite and halloysite. The characteristic spectrum of xylan can be seen in Figure 1a. The absorption band at 3312 cm^−1^ is associated with the stretching of OH groups [34]. The stretching of uronic acid and CH_2_ groups are represented by the bands at 1464 and 1425 cm^−1^, respectively [34,35]. The presence of acetyl groups is given by the absorption bands at 1731 and 1247 cm^−1^ [35]. The peak at 1160 cm^−1^ corresponds to arabinose side chains. The peak at 1634 cm^−1^ is due to absorbed moisture, which shows the hygroscopic nature of xylan [36,37].

In Figure 1a, the typical absorption bands of alginate can be seen at 3280 and 2931 cm^−1^ which can be attributed to OH and aliphatic C–H stretching, respectively [38]. The absorption bands at 1598 and 1411 cm^−1^ can be attributed to the asymmetric and symmetric stretching vibrations of the carboxylate salt of alginate, respectively [38].

Figure 1b shows the FTIR spectra of bentonite and halloysite. As expected, given that they share the same functional groups, both spectra display similar absorption bands. The spectra have distinctive absorption bands at 3619 and 3404 cm^−1^ which are representative of OH groups that are complexed to metals within the clay and those that are not complexed, respectively [39]. The absorption band at 1634 cm^−1^ is due to water absorbed by the clay.

Figure 1c,d shows the FTIR spectra of the xylan and alginate blend films containing bentonite and halloysite, respectively. These spectra show absorption bands that are a mix of the absorption bands seen for xylan and alginate. The absorption bands for bentonite and halloysite are not seen most likely due to the low concentration of the clay fillers. Furthermore, the absorption bands of the clays overlap with those of xylan and alginate. The bands at 3291 and 2933 cm^−1^ represent the OH and CH groups in the samples, respectively. The slight shift for –OH groups from 3312 cm^−1^ (for xylan) and 3280 cm^−1^ (for alginate) to 3291 cm^−1^ may be related to the interaction between the components via hydrogen bonding [40,41]. Furthermore, there is a shift of the asymmetric shift of the carboxylate salt of alginate from 1598cm^−1^ (alginate) to 1595 cm^−1^ (composites), which suggests the disruption of the original hydrogen bonds of alginate, possibly due to the formation of new hydrogen bonds [40,41]. All the components that make up the films consist of a large number of hydroxyl groups, which suggests that the components may possibly bonded to each other through hydrogen bonds. The band at 1595 cm^−1^ is due to the vibration of the carboxylate salt of alginate.

### 3.2. XRD of Xylan–Alginate Composite Films

Figure 2 shows the XRD diffractograms of bentonite, halloysite, and xylan–alginate films containing bentonite and halloysite. The diffraction peaks of bentonite (Figure 2a) at 7.02°, 20.18°, 28.97°, and 62° show the (001), (003), (004), and (060) planes of montmorillonite and confirm that montmorillonite is the major constituent of bentonite [42]. The diffraction peak at 40.34° suggests that there is calcite within the bentonite, whereas the diffraction peak at 54.42° shows that there is quartz present in the bentonite [42].

The characteristic diffraction pattern of halloysite can be seen in Figure 2b. The diffraction peaks at 8.8° and 12.6° are for the (001) facet of the hydrated and dehydrated halloysite, respectively [43]. The characteristic peaks of halloysite can be seen at 20.18°, 25.10°, 35.14°, 38.41°, 55.23°, and 62.37° which correspond to the (100), (002), (110), (003), (210), and (300) planes, respectively, as per JCPDS file 029-1487 [44,45].

The diffractograms of the xylan–alginate films containing bentonite (Figure 2c) show a single diffraction peak which occurs at 19.62. The intensity of this peak increases as the content of bentonite incorporated into the xylan–alginate polymer matrix increases. This is a shift to a lower diffraction angle than that of the pure bentonite clay, and this indicates that the basal spacing has increased from 4.38 to 4.52 Å ((003) diffraction peak). This shows that the space between the clay interlayers has increased when the clay is added to the xylan–alginate matrix. When the polymer chains are intercalated with the clay interlayers, there is an increase in the interlayer spacing leading to a decrease in the diffraction angle [46]. This increase in d-spacing suggests that the polymer chains are intercalated with the clay to a certain extent.

The diffractograms of the films reinforced with halloysite are shown in Figure 2d. Diffraction peaks can be seen at 12.26° and 24.76° which corresponds to the (001) and (002) planes, respectively. The basal spacing increased from 7.03 to 7.21 Å ((001) diffraction peak) for the pure halloysite and the halloysite within the xylan–alginate matrix, respectively. These results imply that the polymer chains are within the clay interlayer, thereby increasing the distance between the interlayer. This would mean that the polymer and the clay are intercalated.

The process of intercalation is shown in Scheme 1. The intercalation of the polymer with the clay results in a well-ordered structure and good interfacial bonding between the polymer and the clay [47]. Intercalation can improve the mechanical and gas barrier properties of the composites.

### 3.3. Colorimetric Analysis

The color of the food packaging material is important since it can affect consumer acceptance of the product [29]. Table 2 shows the color parameters of films containing either bentonite or halloysite. The addition of either bentonite or halloysite to the films results in the L* value of the films decreasing slightly, which means that the films became darker in the presence of clay. The increasing darkness of the films with increasing clay filler content may be due to the clay particles being darker than the xylan–alginate matrix. Furthermore, the films became less green and more yellow as the content of clay in the films increased, as indicated by the a* and b* values increasing with an increase in the clay content. These changes in color may be caused by the difference in color between the xylan–alginate matrix and the clays used. Another reason for the change in the color of the films may be that there are electrostatic interactions between the clay and the xylan–alginate matrix [48]. Johansson et al. studied these changes in color with the addition of clay into the polymer matrix, and attributed the changes in color to the leaching of metal ions from the clay into the polymer matrix [22]. Similar results were also reported by Alipoormazandarani et al. and Saurabh et al. when adding halloysite and bentonite into to soybean polysaccharides and guar gum, respectively [49,50]. In terms of color value, the films containing bentonite are darker compared to films containing halloysite. The difference in color, particle size, and particle shape between the bentonite and halloysite may be the source of the bentonite making the films darker than the halloysite [51,52]. Compared to low density polyethylene (LDPE), which is traditionally used for food packaging, the xylan–alginate films are somewhat darker.

### 3.4. Light Transmittance of Films

Figure 3 shows the UV–vis transmittance spectra of films containing bentonite and halloysite. Light transmittance is an important parameter of food packaging films, influencing the appearance and attractiveness of the films as well as their ability to protect products from degradation due to light exposure [54]. Films used as packaging for fresh foods such as meat, fruits, and vegetables should have a high light transmittance in the visible region so that consumers may have a realistic view of the product [54,55]. However, foods that may degrade upon being exposed to either visible light or UV light require packaging materials that are opaque [54].

The light transmittance spectra of the xylan–alginate films containing bentonite can be divided into three regions, viz., (i) 800–550 nm, (ii) 550–380 nm, and (iii) 380–200 nm (Figure 3a). In the region of 800–550 nm, all films containing bentonite display similar light transmittance as the neat film. In the 550–380 nm region, there is a reduction in visible light transmittance with the addition of 1 wt % of bentonite to the xylan–alginate matrix, and further addition of bentonite results in an increase in transmittance. This could be due to an increase in clay dispersion with an increase in the bentonite content. Light transmittance decreases with an increase in the size of particles within the polymer matrix; therefore, the agglomeration of particles leads to lower light transmittance [14,52]. In contrast, Monteiro et al. observed that the incorporation of bentonite into starch films reduced the light transmittance of the films, which was ascribed to an agglomeration of bentonite [56]. In the UV region (380–200 nm), the light transmittance decreases upon the addition of 1 wt % of bentonite, but then increases upon further addition of bentonite. This may also be due to an increase in clay particle dispersion with an increase in clay content. However, for all films containing bentonite, there was lower UV transmittance compared to the neat film.

In the case of halloysite, the films displayed a concentration dependent increase in the amount of light absorbed in both the visible and the UV regions (Figure 3b). This increase in light absorbance may be due to the agglomeration of the halloysite particles within the xylan–alginate matrix. Similar observations were made by Lee et al. and He et al. who found that light transmittance decreased with the addition of halloysite to chitosan and starch films, respectively [29,57]. The addition of clay, both bentonite and halloysite, decreases the amount of transmittance of the films in the UV region of the spectra. The reduction of UV transmittance can be regarded as an improvement on the properties of the films as this would reduce the risk of lipid oxidation caused by UV radiation [58]. Films containing bentonite would be suitable for packaging applications regarding fresh foods, as they are highly transparent in the visible region and can provide additional protection from degradation caused by UV radiation. Films reinforced with halloysite on the other hand would be appropriate for packaging of foods that degrade when exposed to visible light, such as products containing chlorophylls [51].

### 3.5. Mechanical Properties

High tensile strength and elongation at break are properties that are desirable for food packaging applications as the packaging must be able to withstand handling during transportation of the food. Table 3 shows the mechanical properties of the films containing bentonite and halloysite. The addition of bentonite to the films causes a linear increase in the tensile strength and Young’s modulus of the samples, compared to the control film. The tensile strength and Young’s modulus of the film containing 5 wt % of bentonite showed an increase of 112.62% and 76.25%, respectively, compared to the control film. This shows that the bentonite acted as an effective reinforcement phase for the xylan–alginate matrix. This effective reinforcement may be due to good dispersion within the polymer matrix combined with the strain-induced alignment of the clay particles with the polymer chains. Furthermore, this increase in tensile strength may be due to intercalation of the clay and polymers as can be seen in Figure 2b. The elongation at break of the films containing bentonite are very similar to the control film. Similar observations have been reported by other authors such as Ortiz-Zarama et al. and Moraes et al. who reported that the addition of bentonite to gelatin and starch films increased the tensile strength and decreased the elongation at break of the films [20,21].

The addition of 1 wt % halloysite to the xylan–alginate matrix increases the tensile strength and Young’s modulus compared to the control film. The tensile strength was increased by 24.97% and the Young’s modulus was increased by 10.87%, however, further increase in halloysite content (3 wt % and 5 wt %) led to a reduction in both tensile strength and modulus. This may be attributed to the agglomeration of halloysite at content higher than 1 wt %, which limits the load transfer from matrix to halloysite, hence acting as defect sites which will cause failure under tension [46]. A similar behavior was reported by Gorrasi et al. in which the addition of more than 10 wt % of halloysite to pectin films resulted in a decrease in the tensile strength of the films, which was attributed to aggregation of the halloysite [27]. Lee et al. also noted a decrease in tensile strength when the halloysite concentration exceeded 10 wt % in chitosan films [29]. For the xylan–alginate matrix used in this study, the halloysite begins to agglomerate and decrease tensile strength at lower loadings than in the previous studies. The hydrogen bonding between the halloysite particles may be stronger than the hydrogen bonding that occurs between the halloysite and the xylan–alginate matrix, leading the clay to agglomerate. The films containing halloysite have similar elongation at break values as the control film. The addition of halloysite beyond 1 wt % to the films results in weaker and more brittle films. The elongation at break of the xylan–alginate films is much lower than the elongation at break shown for LDPE; however, the stress at break is equivalent or superior. This suggests that in applications that do not require a high elongation at break, the xylan–alginate film containing 5 wt % bentonite may be a suitable replacement.

### 3.6. Thermogravimetric Analysis

Figure 4 shows the derivative thermograms (DTGA) of the alginate, bentonite, halloysite, and xylan as well as films containing bentonite and halloysite. The respective thermograms of the DTGA are provided in the Appendix A section as Appendix A. In Figure 4a, it can be seen that bentonite displayed a single degradation step at 400 °C which can be associated with the loss of hydroxyl groups on the surface of the clay and the loss of interlayer water [60]. Alginate, halloysite, and xylan (Figure 4a) all display a minor weight loss that is associated with the evaporation of absorbed moisture which occurs below 150 °C. Xylan displays a single degradation step with the onset of thermal degradation (T_onset_) taking place at 170 °C and maximum degradation (T_max_) occurring at 187 °C. Alginate also displayed a single degradation step, with the T_onset_ and T_max_ of alginate occurring at 211 °C and 258 °C, respectively. Xylan and alginate follow similar thermal degradation mechanisms, and the T_onset_ signals the loss of functional groups and the breakage of C-O-C glycosidic bonds. This is followed by the release of carbon dioxide and water vapor. When alginate thermally degrades, Na_2_CO_3_ forms, resulting in a higher char yield (45%) compared to xylan (31%).

The xylan–alginate films incorporated with either bentonite or halloysite show four distinctive thermal events (Figure 4b,c), which are summarized in Table 4. The first thermal event occurs at 86–98 °C and is associated with the evaporation of moisture absorbed by the films. The second thermal event is attributed to the degradation of xylan and occurs between 173–175 °C (i.e., the T_max_ of the nanocomposite samples), while the T_onset_ of the samples takes place at ~152 °C. The third thermal event occurs 184 °C and is related to the degradation of alginate in the sample. The fourth thermal event that occurs is the evaporation of the glycerol in the films at 227 °C [61]. The T_onset_ of the blends occurs at a lower temperature than the pure polymers. This is due to the addition of glycerol which reduces intermolecular bonding and allows the samples to be thermally degraded more easily. Khan et al. made a similar observation on the reduction of thermal stability of starch films upon addition of glycerol [62]. The inclusion of either clay did not affect the T_onset_ of the blends. However, the rate of T_max_ of the samples is reduced by the incorporation of clay. These results show that the amount of clay in the films does not result in the films being more thermally stable as the onset of temperature for thermal degradation remains the same regardless of clay content. However, the rate of weight loss during T_max_ is reduced.

For the samples containing bentonite, the char yield increased from 34% to 38% as the amount of bentonite in the sample increased. The char yield for the samples containing halloysite increased from 34% for the control film to 37% for all the films containing halloysite.

### 3.7. Water Vapor Permeability and Solubility

Food packaging materials should ideally insulate the food enclosed within it from odors, loss of flavor, chemicals, oxygen transmittance, and water vapor permeability [63]. Amongst these, the water vapor permeability (WVP) is of significant importance with regard to the control of moisture transferred between the food and the exterior environment. A high WVP can lead to the microbial spoilage of the food. Table 5 shows the WVP of the films containing either bentonite or halloysite. The addition of either clay reduces the WVP of the films compared to the control film. For either clay, the WVP of the films decreases with the incorporation of clay in a concentration dependent manner. The WVP of the control film was found to be 3.94 × 10^−10^ g·s^−1^·m^−1^·Pa^−1^ and this was reduced to 2.01 × 10^−10^ g·s^−1^·m^−1^·Pa^−1^ with the addition of 5 wt % of either clay. This represents a reduction of 48.98% in the WVP of the films with the incorporation of 5 wt % of clay. A major constituent of both clays is silica, and silica particles are impermeable to water vapor. This means that the incorporation of clay into the xylan–alginate matrix creates a more complex path for water molecules to travel through and thereby reduces the amount of water that can travel through the film as compared to unfilled xylan–alginate matrix. Scheme 2 illustrates the more complex pathway created by the incorporation of clay particles into the xylan–alginate matrix. The pathway for water molecules becomes more complex/longer with an increasing amount of clay incorporated into the xylan–alginate matrix. The longer the pathway created for the gas molecules to travel through, the greater the reduction in the rate of gas diffusion. Alboofetileh et al. reported that the incorporation of montmorillonite (MMT) into alginate films reduced the WVP of the films, and this was ascribed to the impermeable silicate layers of montmorillonite [46]. When the loading of either bentonite or halloysite is the same, the films show very similar WVP properties. These results show that clay nanofillers can be used to reduce the WVP of xylan–alginate blends and that there is very little difference between bentonite and halloysite and their effect on WVP properties of xylan–alginate films. Similar results were reported by Lee et al. who noted that the addition of 30 wt % halloysite to chitosan films resulted in a decrease of WVP by 14.18% [29]. A study by Monteriro et al. indicated that the incorporation of bentonite into cassava starch resulted in a decrease of the WVP of the films [64].

In food packaging applications, where the food needs to be protected from water or the packaging must maintain its mechanical properties in an aqueous environment, low solubility of the packaging material is desirable [65]. However, in other applications, such as the encapsulation of food, soluble sachets, or for edible films, high solubility is desirable [55,66]. The water solubility of xylan–alginate films containing either bentonite or halloysite is given in Table 5. The films containing bentonite and halloysite show very similar solubilities compared to the control film. This shows that xylan and alginate are both highly sensitive to water and that the addition of clay to these films does not decrease the sensitivity of the polymers to water. These results are in contrast with the results reported by Abdollahi et al. who reported that the solubility of alginate decreased with the addition of MMT from 100% to 61.35% [65]. Previously, Yang et al. reported a decrease in solubility of alginate films with the incorporation of silica and attributed this to strong hydrogen bonding between alginate and silica [14]. The xylan–alginate matrix and the clays can interact through hydrogen bonding due to the presence of hydroxyl groups in all the constituents. It could be possible that, in this study, the hydrogen bonding strength between the xylan–alginate matrix and both clays is weaker than the binding energy of water to the xylan and alginate. The high solubility of the xylan–alginate films incorporated with bentonite and halloysite indicates that these films would be good for use as edible films and/or soluble sachets.

### 3.8. Water Sorption

The moisture content of biopolymer films influences the mechanical and barrier properties as well as shelf life of biopolymer films. Absorbed water molecules expand the distance between polymer chains, reducing Young’s modulus and tensile strength while increasing the elongation at break of the films. This increase in free volume also has a negative influence on the gas barrier properties of the films [67]. The water sorption isotherms of xylan–alginate films containing either bentonite or halloysite are shown in Figure 5. The amount of moisture absorbed by the films increases as relative humidity the films are exposed to increases. All samples exhibited type III isotherms according to the BET classification of isotherms, as the curvature of the isotherm is convex toward the *a_w_* axis over the entire range [68]. This suggests that the bonding between the water molecules and the sorption sites (hydroxyl groups) is weaker than the bonding between water molecules [68]. The films display a two-step process of moisture absorption. The first step is the formation of a water monolayer where water molecules bond to surface hydroxyl groups. This occurs below 0.33 *a_w_* for the control film and below 0.11 *a_w_* for the films reinforced with clay. The second step is the multilayer absorption of water molecules, when the absorbed water in the monolayer provides new sorption sites for further absorption of water causing a sharp increase in moisture content. The second step occurs above 0.33 and 0.11 *a_w_* for the control film and the composite films, respectively. These results show that the incorporation of clay into xylan–alginate matrix causes the water monolayer to occur at lower *a_w_*, compared to the neat polymer. The clay nanoparticles also have a large number of surface hydroxyl groups, which results in a greater hydroxyl group availability for water molecules to bind to. A greater availability of hydroxyl groups may allow for the water monolayer to form at lower *a_w_*. At 0.33 and 0.54 *a_w_*, the films reinforced with clay show a greater moisture content than the neat film, however, at 0.75 and 0.96 *a_w_*, all films had a similar moisture content. Due to the films containing clay forming a water monolayer at lower *a_w_*, the multilayer absorption also occurs at lower *a_w_* which results in a higher moisture content for the films containing clay at 0.33 and 0.54 *a_w_*.

The physical constants of moisture sorption as calculated using the Guggenheim–Anderson–de Boer (GAB) model are shown in Table 6. The C value is an indication of the bond strength between the water molecules forming the monolayer and the surface hydroxyl groups [69]. All samples exhibited very similar C values which shows that the incorporation of clay did not influence the bond strength between water and surface hydroxyl groups. The W_m_ value is the moisture content of the water monolayer [69]. If the W_m_ value is ≥10 wt %, the food product will have reduced physical and chemical stability [70]. All films exhibited a water monolayer content of <1 wt %, demonstrating that all films would be physically and chemically stable. The water monolayer content shows a similar trend with the addition of either clay. The water monolayer content increases in order of control < 3 Ben < 1 Ben < 5 Ben and control < 3 Hal < 5 Hal < 1 Hal. The water monolayer content is directly related to the number of binding sites, which would be hydroxyl groups for xylan–alginate films [71]. This implies that when the clay content is 3 wt %, the hydrogen bonding between the clay and the xylan–alginate matrix is strongest as this would reduce the number of available hydroxyl groups for water to bond with. The K value is the multilayer absorption factor and should be between 0.24 ≤ K ≤ 1, and the C value should be in the range of 5.67 ≤ C ≤ ∞ [69,72]. If the K and C values do not fall within the recommended range, the true monolayer content may vary by more than 15.5% of the calculated amount. While the K value of all samples fell within the recommended range, the C value was outside of the recommended range. This means that the true monolayer content may vary by more than 15.5% of the calculated amount. Given that the calculated monolayer content was well below 10 wt %, this should not have an influence on the chemical and physical stability of the films. All the xylan–alginate films reinforced with clay displayed water monolayer contents in the recommended range suggesting that these films would be suitable as food packaging materials and/or edible films.

### 3.9. Scanning Electron Microscopy

Figure 6a–c shows scanning electron microscopy (SEM) images of the cross-sectional surfaces of xylan–alginate films and films containing 5 wt % of halloysite and bentonite, respectively. The images of the nanocomposite films for films containing nanoclays were rougher than that of the neat xylan–alginate film. It was also observed that the cross-sectional surface become rougher with an increase in nanoclay loading (the SEM images of 1 Ben, 3 Ben, 1 Hal, and 3 Hal are shown in Appendix A). In the case of halloysite-based nanocomposite films, smooth surfaces with white spots (highlighted in Figure 6b) within the cavities of the films which could be associated with halloysite clay particles were observed. On the other hand, the bentonite-based composite films exhibited a more compact structure than the neat film with the presence of white spots (highlighted) as well as pull-outs (marked with arrows) (Figure 6c). It can be observed that the white spots are evenly distributed indicating that bentonite was well-dispersed within the xylan–alginate matrix even at higher concentrations. These images provide insight regarding the mechanical properties seen in Section 3.5 as the more compact structure of the bentonite-reinforced films resulted in better mechanical properties.

## 4. Conclusions

In this study, nanocomposite films were prepared by incorporating either bentonite or halloysite into a polymer matrix of xylan and alginate using the solvent casting method. The color parameters of the films were tested, and it was observed that the incorporation of both bentonite and halloysite resulted in the films being darker. The presence of bentonite or halloysite also decreased the UV transmittance of the films which would be beneficial for food packaging due to decreased lipid oxidation of food. The incorporation of bentonite improved the tensile strength and Young’s modulus of the films compared to the control film due to effective stress transfer from the polymer matrix to the clay particles. The addition of 1 wt% of halloysite caused an increase in the tensile strength and Young’s modulus of the films. However, at higher loading levels, the tensile strength decreased. This was attributed to the agglomeration of halloysite which acted as defect sites for the composites. The incorporation of either clay resulted in a large decrease in the WVP of the films due to the clay particles being impermeable to water molecules.

Bentonite clay was a better reinforcement material than halloysite for the xylan–alginate polymer matrix as the bentonite nanocomposites showed better optical and mechanical properties than the halloysite nanocomposites.

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
