# Peer review of "Effect of Clay Nanofillers on the Mechanical and Water Vapor Permeability Properties of Xylan–Alginate Films"

_polymers, 2020, doi:10.3390/polym12102279_

Round 1

Reviewer 1 Report

In this study, xylan-alginate based films were reinforced with nanoclays (bentonite or halloysite) by solvent casting technique. The effect of the nanoclay loadings on various properties (mechanical, optical, thermal, solubility, water sorption and WVP of the xylan-alginate films) were examined for their application as food packaging materials. However, the incorporation of bentonite or halloysite in polymeric matrix is a common method for the modification of films and there are numerous researches reported on this topic.  In addition, much work needs to do. Some suggestions are given as follows:

1. Introduction

The incorporation of bentonite or halloysite in polymeric matrix is a common method for the modification of films and there are numerous researches reported on this topic. The novelty of this work should be clearly addressed.

2. Materials and Methods

  • L76, the detailed information of bentonite and halloysite (physical and chemical characteristic parameter) should be provided.
  • Statistical analysis is necessary.
  1. Results and discussion
  • The quality of Figure 1 is poor. In XRD, please ascribe the peaks more delicately with miller indices.
  • L211-212, what change in absorption bands can indicate the formation of hydrogen bonds?
  • In XRD, the patterns 1Ben and 1Hal are significantly different from those of (3Ben, 5Ben) and (3Hal, 5 Hal), what is the reason?
  • In Colorimetric analysis, the colour parameters of pure bentonite or halloysite are missing.
  • In Figure 3 a, it is hard to understand that the transmittance of 5Ben is higher than that of 1Ben or 3Ben.
  • As indicated from Table3, both the tensile strength and the elongation at break of the films increased with the addition of bentonite or halloysite, an reasonable explanation is needed.
  • In Figure 4, the T1-T4 should be indicated. In addition, the meanings of T1-T4 need definition.
  • What about the barrier of the films to gas such as O2 or CO2?
  • In Figure 6, (1Ben, 3Ben) and (1Hal, 3 Hal) are missing.

Author Response

Introduction

This is the first study to incorporate bentonite and halloysite into a matrix of xylan-alginate, also it is the first study to directly compare the efficacy of the two reinforcement materials on the properties of materials intended for food packaging.

Experimental

Other than the FT-IR, XRD and TGA presented in the manuscript only surface area analysis was done on the bentonite and halloysite.

Results

  • The miller indices have been added to Figure 2.
  • L211-212. There is no FT-IR evidence to support the claim of hydrogen bonding however, due hydroxyl groups being present on all components it is postulated that the components interact through hydrogen bonding.
  • The quantity of bentonite and halloysite incorporated into the composites is very small this means that the diffraction peaks will be very weak and not all the peaks will be represented in the samples. For the samples containing 1wt% of clay reinforcement there is not enough clay present in the samples to produce XRD diffraction peaks.
  • Unfortunately, the colour parameters of pure bentonite and halloysite was not measured, only the effect of the incorporation of bentonite and halloysite on the colour parameters of the xylan-alginate films.
  • The transmittance of the films incorporated with bentonite increases in the order 1 Ben<3 Ben<5 Ben. The transmittance of a plastic composite increases as the dispersion of the reinforcement increases. The hypothesis for the phenomena observed in this manuscript is that the dispersion of the bentonite increases with an increase in bentonite content.
  • It is possible that due to the good dispersion of the bentonite with in the polymer matrix that both the elongation at break and the tensile strength increase with increasing bentonite content. This could also be caused by the effective transfer of stress from the polymer matrix to the reinforcement phase, which may allow the polymer chains to extend further without breaking.
  • The definition of T1, T2, T3 and T4 have been added below Table 4. And the positions of T1, T2, T3 and T4 have been added to Figure 4 a and b.
  • The barrier properties of the films towards oxygen and carbon dioxide was not tested as water vapour permeability of biocomposites is a generally recognized issue and this property was considered to be of greater importance.
  • The SEM images of 1 Ben, 3 Ben, 1Hal and 3Hal were submitted in the supplementary information.

Reviewer 2 Report

the paper concerns the preparation and characterization of xylan-alginate composites.

the paper is quite interesting: to help the reader and improve the interest for the paper some suggestions:

1) a simple scheme of the chemical structure of the components could be added

2)as for the properties which are detected a reference value typical for packaging materials or a comparison could be useful to be added

3) in the figures the peaks considered to study in particular the dspacing could be indicated

4) apparently there is a different interaction between the matrix and bentonite or halloysite: I did not expect a Thermal behaviour of the composites so similar. have the authors a comment? surely a deeper inestigation of the nature of the interaction coul be interesting.

5) A curiosity: did the authors experiments with ratios different from 1 to 1 between the two components of the matrix ?

Author Response

  • The chemical structures of xylan and alginate have been added to the supplementary information document as Figure S1.
  • Basal spacing and d-spacing are both calculated in the using Bragg’s equation. Basal spacing standing for the interlayer space between two clay layers and d-spacing the space between two layers of atoms in a crystalline structure.
  • Where possible the properties of LDPE have been added for comparison.
  • Both bentonite and halloysite interact with the xylan-alginate matrix through intercalation.
  • Only the ratios of 1, 3 and 5 wt% of bentonite and halloysite incorporation were tested.

Reviewer 3 Report

The authors presented their studies about Effect of clay nanofillers on the mechanical and water vapour permeability properties of xylan-alginate films.

They showed a extensively research about their material.

They subjected their material to a series of tests and proved that the optical and material proerties are improved when the nanoclay particles are added to xylan-alginate films. The nanoclay particles did not influenced on thermal stability and solubility of the nanocomposite films and decreased the UV transmittance of the films which would be beneficial for food packaging.

This paper could be approved.

Author Response

Thank you for reviewing this manuscript.

Round 2

Reviewer 1 Report

  1. There is no FT-IR evidence to support the claim of hydrogen bonding, why?
  2. Figure 2c, the peaks of 003 were not present in 3Ben and 1Ben, but it was present in the control, why?
  3. For colorimetric analysis, the color parameters of pure bentonite or halloysite should be measured.
  4. In Figure 3 a, it is hard to understand that the transmittance of 5Ben is higher than that of 1Ben or 3Ben. That is to say, the incorporation of the inorganic particles of bentonite can enhance the transmittance of the composite film, why?
  5. As the authors stated both the elongation at break and the tensile strength increase with increasing bentonite content. This could also be caused by the effective transfer of stress from the polymer matrix to the reinforcement phase, which may allow the polymer chains to extend further without breaking. Why not try the contents of bentonite or halloysite above 5wt%?
  6. The gas barrier properties of the films were also of greater importance for food packaging.

Author Response

Comment: There is no FT-IR evidence to support the claim of hydrogen bonding, why?

Response: Based on your comment we have modified the sentence (please see L212). Our claim stems from the presence of a large number of hydroxyl groups of all components which may result in hydrogen bonding between these components.

Comment: Figure 2c, the peaks of 003 were not present in 3Ben and 1Ben, but it was present in the control, why?

Response: We have modified the sentence indicating that the peak at 19.62° is directly dependent on the concentration of Bentonite (L236). Please also see black line in Figure 2c (which represents the control) and does not show a peak at 19.62°.

Comment: For colorimetric analysis, the color parameters of pure bentonite or halloysite should be measured.

Response: The reviewer’s comment is noted. We will consider doing such tests in future for comparison purposes.

Comment:  In Figure 3 a, it is hard to understand that the transmittance of 5Ben is higher than that of 1Ben or 3Ben. That is to say, the incorporation of the inorganic particles of bentonite can enhance the transmittance of the composite film, why?

Response: Transmittance increases as the amount of bentonite incorporated increases implying this is a trend, however the transmittance of all samples containing bentonite is lower than the control sample. Transmittance increases as the particle size decreases, the SEM (Figure 6) shows that halloysite agglomerates (Figure 6b) while the bentonite does not (Figure 6c). It is possible that at higher concentrations there is increased collisions between the clay particles during mixing which may led to reduced particle size. This would in turn increase the dispersion of the bentonite particles.

Comment: As the authors stated both the elongation at break and the tensile strength increase with increasing bentonite content. This could also be caused by the effective transfer of stress from the polymer matrix to the reinforcement phase, which may allow the polymer chains to extend further without breaking. Why not try the contents of bentonite or halloysite above 5wt%?

Response: Thank you for such valuable input, future studies will include evaluation of effect of high loading of clay on mechanical properties of xylan-alginate blends.

Comment:  The gas barrier properties of the films were also of greater importance for food packaging.

Response: It is widely acknowledged that the main disadvantage of using biopolymers is that they have a very high-water vapour permeability which is detrimental to food storage lifetime. Therefore, most studies dealing with biopolymers intended for food packaging applications only test for water vapour permeability. Furthermore, the availability of instruments to test for oxygen and/or carbon dioxide permeability is rather limited at my location and the cup method for WVP testing is a readily available technique.

Round 3

Reviewer 1 Report

It may be at present although there is much left to do.